# Reducing Barriers for Best Practice in People Living with Dementia: Cross-Cultural Adaptation and Content Validity of the Brazilian Version of the Pain Assessment in Impaired Cognition (PAIC-15) Meta-Tool

**DOI:** 10.3390/ijerph22091324

**Published:** 2025-08-26

**Authors:** Paula Schmidt Azevedo, Juli Thomaz de Souza, Alessandro Ferrari Jacinto, Déborah Oliveira, Fania Cristina dos Santos, Marcos Ferreira Minicucci, Paulo José Fortes Villas Boas, Wilco Achterberg, Patrick Alexander Wachholz

**Affiliations:** 1Internal Medicine Department, Botucatu Medical School, São Paulo State University (UNESP), Botucatu 18618-687, Brazil; schmidt.azevedo@unesp.br (P.S.A.); marcos.minicucci@unesp.br (M.F.M.); paulo.boas@unesp.br (P.J.F.V.B.); patrick.wachholz@unesp.br (P.A.W.); 2Department of Medicine, Centro Universitário de Adamantina (FAI), Adamantina 17800-000, Brazil; alessandrojacinto@fai.com.br; 3Faculty of Nursing, Campus Viña del Mar, Universidad Andrés Bello, Viña del Mar 2531015, Chile; deborah.deoliveira@unab.cl; 4Millenium Institute for Care Research (MICARE), Santiago 8370134, Chile; 5Department of Medicine, Paulista School of Medicine, Federal University of Sao Paulo (UNIFESP), São Paulo 04025-002, Brazil; cristina.fania@unifesp.br; 6Department of Public Health and Primary Care, Leiden University Medical Center, 2333 ZA Leiden, The Netherlands; w.p.achterberg@lumc.nl

**Keywords:** impaired cognition, dementia, pain assessment, tool, Brazilian Portuguese

## Abstract

Background: Pain detection and management in older people with impaired cognition is an unmet need, leading to healthcare inequalities. This study aimed to enhance pain detection by translating and culturally adapting the PAIC-15 questionnaire into Brazilian Portuguese, establishing its content validity and comparing findings with a similar Dutch study. Methods: Following the same international standards used in the English-to-Dutch validation, we translated and culturally adapted PAIC-15 for Brazil. Content validity was assessed using two questions: (A) whether the item indicated pain and (B) whether it was specific to pain. An online form was distributed to ‘target users’ (nurses and geriatricians in LTCFs or other settings) and other healthcare professionals (HCPs) experienced in pain assessment in people with cognitive decline. Results were compared with the English-to-Dutch validation. Results: A total of 103 HCPs responded (mean age 45.2 ± 11.4 years)—76 (74%) were female and 54 (52%) had previously used pain assessment tools. Among the 63 target users, 13 items (87%) were validated as indicative and/or specific to pain. Comparing Brazilian and Dutch users, 10 items (67%) showed agreement on pain specificity. Conclusions: The Brazilian Portuguese PAIC-15 is now available, showing acceptable content validity. Linguistic challenges highlight the need for training and further validation in larger samples.

## 1. Introduction

Pain is an individual experience of unpleasant sensory and emotional sensations associated with (or resembling that associated with) actual or potential tissue damage [1]. Pain is often neglected and undertreated in older people and can be influenced and related to varying degrees of biological, psychological, and social factors [2].

Managing pain can be challenging in people with communication impairment, such as in people living with moderate to severe dementia [3]. They often experience concomitant health conditions that contribute to significant declines in functional ability and can be associated with acute pain after adverse or striking life events (i.e., falls, fractures, and surgeries), complications of underlying conditions (i.e., rheumatological, orthopedic, or other organic dysfunctions), and chronic pain (e.g., low back pain, osteoarthritis [3].

Establishing the prevalence of pain in people living with dementia (PLWD) is necessary but can be challenging. The evidence is imprecise due to difficulties distinguishing expressions and behaviors related to pain from those related to other experiences (such as fear, annoyance, or hunger). Yet, nursing home residents with dementia have less frequent pain assessments and less non-pharmacological or pharmacological treatment compared with those with preserved cognition [4,5]. In addition, advanced dementia clinical pictures may lead to more severe pain. Therefore, insufficient pain assessment and management in PLWD leads to inequalities in the healthcare pathways compared to young adults and with older people with preserved cognition, and should be understood as a threat to their human rights and disability discrimination act [6,7,8]. Despite self-reporting being considered the “gold standard” of pain assessment, this approach is insufficient for PLWD [9]. In 2006, the American Society for Pain Management Nursing published a position statement and clinical practice recommendations for pain assessment in nonverbal patients, recommending a comprehensive, hierarchical approach that integrates self-report and observations of pain [10].

Accurate pain assessments are essential to ensure effective and person-centered treatment for PLWD. Considering its importance, research initiatives and consortia have been dedicated to developing instruments and scales to measure pain in older people with cognitive impairment. Although several observational and informant-based assessment tools have been designed to identify specific behaviors, a significant disparity exists in their operationalization methods [11]. Existing tools frequently need more comprehensive data on face and construct validity, reliability, and responsiveness [12]. A systematic review of pain assessment found many available tools for cognitively impaired populations. However, more evidence is needed about their reliability and clinical utility. Based on this review, no single tool was recommended for clinical practice; however, a structured and validated tool is essential for increasing the detection of pain [11].

The initial version of the Pain Assessment in Impaired Cognition (PAIC) tool was developed by an international panel of experts and clinical researchers from the European Cooperation in Science and Technology initiative. It was based on existing instruments embedded in the practicalities of clinical practice and user-based design. It identified the most meaningful pain assessment items based on empirical value to develop a robust evaluation and consensus procedure, creating a final tool and toolkit with universal utility. PAIC had 36 items reflecting three domains: facial expressions, vocalization, and body movements [12,13]. Then, the items were reduced to 15, forming the PAIC-15, which was divided equally among the three domains and selected from 13 previous studies in 8 European countries that tested the psychometric properties in 587 individuals with dementia and other populations [13].

The PAIC-15 [13] has been translated into various languages, including Spanish, Dutch, German, Japanese, and French, and is available on the tool’s dedicated webpage (https://paic15.com/en/start-en/, accessed on 5 August 2025). However, no Brazilian Portuguese version has been available to date. The interest in having a Brazilian Portuguese version of PAIC-15 encompasses the characteristics abovementioned, such as the balance between the domains and the broadly studied psychometric properties originating from a meta-tool. This study aimed to translate and culturally adapt the original English version of the PAIC-15 for use in Brazil, establish its content validity, and compare the findings with those of a previous similar study developed in the Netherlands. Having a structured, multidomain tool validated in different languages promotes the awareness of the importance of assessing pain and pain-related behaviors. Therefore, it increases the likelihood of adequate pain management, including for individuals with impaired cognition.

## 2. Materials and Methods

### 2.1. Translation and Cultural Adaptation

The PAIC-15 translation and cultural adaptation were undertaken in line with the Guidelines for Establishing Cultural Equivalency of Instruments [14], replicating the work of van Dalen-kok et al. [15]. The original English version was submitted to forward translation into Brazilian Portuguese by two direct and independent translators whose native language is Brazilian Portuguese. One of the translators is a healthcare professional with experience in treating patients with pain; the other translator is a healthcare professional without direct experience with treatments for patients with pain.

Then, a group of six researchers (researcher judges) with experience treating patients with pain and/or dementia who speak Brazilian Portuguese as their native language and English as their second proficient language compared both translations. The group met via videoconference and assessed the tool’s face validity by analyzing the items’ clarity, suitability, and the instrument’s idiomatic, semantic, cultural, and conceptual equivalences.

The first draft of the Brazilian version was back-translated into English by two independent translators without knowledge of the original tool. The first is native to English and has significant Brazilian Portuguese language skills, with experience translating and proofreading health texts. The second one works for a translation company (TranslationPal LLC, Iowa City, IA, USA) without research or clinical experience.

Subsequently, the same group of researcher judges compared and synthesized the two back-translated documents into one document. This second draft in English was sent to the authors of the PAIC-15 tool to compare it with the original instrument.

All discrepancies were discussed and adjusted in a meeting, which led to a newly revised version. This version was then presented to three nurses (nurse judges) with extensive experience caring for older people with cognitive impairment. They were invited to a ‘think aloud,’ whereby they were asked to read and think while verbalizing their views concerning their experiences in assisting PLWD, including considerations on the tool’s comprehension and practical and theoretical relevance in identifying pain-related signs13. Finally, a larger group of HCPs assessed the PAIC-15 Brazilian version (Appendix A) to establish content validity.

### 2.2. Content Validation

#### 2.2.1. Sample

First, we tested the Brazilian version of PAIC-15 with an overall group of HCPs working in long-term care facilities (LTCF) or other relevant settings. The inclusion criteria were HCPs aged 18 and older with a minimum of experience in pain assessment and management, as well as providing care for people with dementia. The exclusion criteria were that the participant did not complete both parts of the questionnaire.

Among these, nurses in LTCF and geriatricians in any setting were considered ‘target users.’ They were analyzed separately from the other HCPs because they were the same as those previously included in the validation process, PAIC-15 from English to Dutch [15]. Therefore, comparing the validation process between two different nations was possible.

#### 2.2.2. Procedures

We replicated the methodology used by van Dalen-Kok et al. in their validation of the PAIC-15 from English to Dutch [15]. We recruited participants via email, social media, cross-platform instant messaging applications, and among the participants of a Brazilian Conference of Geriatrics and Gerontology. The form was answered via an online Google Form^®^ containing a consent form and the study questions.

After giving consent, they were asked five questions about their general characteristics: occupation, age, whether they felt competent to assess pain in persons with dementia, any pain measurement tools, and how often they used them [15]. In addition, they were asked to respond to the two questions about whether they considered each PAIC-15 item as (a) indicative of pain and/or (b) specific to pain [15].

Finally, the participants were invited to provide free commentaries and thoughts and express their opinions and experiences regarding the instrument.

A non-probabilistic sample was used; however, we utilized virtual-based spaces and a national conference to attempt to reach healthcare professionals involved in the care of older adults [16].

### 2.3. Data Analysis

Regarding the question ‘A—whether the item is indicative of pain,’ we used the content validation coefficient (CVC). Responses were scored on a 4-point Likert scale: 1 = No, definitely not; 2 = No, probably not; 3 = Yes, probably; 4 = Yes, definitely. The CVC was calculated considering the average score of all HCPs divided by the maximum score for each item (four). A CVC ≥80% was considered acceptable. The CVC was calculated for the overall group and each group of professionals who responded to the questions [17,18].

To ascertain the question ‘B—whether the item is specific to pain,’ we considered it acceptable if at least 50% of participants responded that the item was specific to pain in cognitive impairment and not to the other condition: depression, delirium, anxiety, or dementia. The participants could choose only one option among the four [15]. Question B was analyzed separately for the overall group and target users. The analysis was performed in Excel Microsoft 365^®^ and the graphics in Graph Pad Prism 6^®^.

We used thematic analysis for the open-ended question. The first step was to review the raw responses and begin the process of preliminary data coding utilizing a combination of deductive codes drawn from the research questions and inductive codes generated by the data. We coded the transcripts in batches and iteratively discussed and established consensus on coding decisions. The research team interpreted the reduced data—including the connections between codes—to develop themes that depicted the reported features of participants’ perceptions of the PAIC-15 tool.

Considering the peculiarities and differences between the health and long-term care systems in Brazil and the Netherlands, we compared the agreement on question B among the Brazilian target users with Dutch nurses and older people care physicians working in LTCF [15]. The option for comparing only question B for the target user was because the healthcare professionals, questions, and analysis methods were similar. For question A, it was slightly different.

## 3. Results

### 3.1. PAIC-15 Translation and Participants’ Characteristics

The translated version of PAIC-15 is shown in Appendix A, including a detailed description of the steps taken to produce it. One hundred and three HCPs from different Brazilian regions responded to the questionnaire. One participant from the non-LTCF geriatrician group did not answer whether the items were considered indicative of pain, so the sample size for Part A was 102.

They were 16 (15.5%) nurses and 23 (22.3%) geriatricians working in LTCF; 24 (23.3%) geriatricians working in other healthcare settings; 7 (6.7%) hospital nurses; 19 (18.4%) other health professionals (physiotherapist, speech therapist, occupational therapist, dentist, caregiver, or not specified); 12 (11.6%) non-geriatrician physicians; and 2 (1.9%) LTCF nursing technicians. The mean age was 45.2 ± 11.4 years; 76 (74%) were female, and 54 (52%) reported using tools to measure pain. Target users summed 63 respondents.

### 3.2. PAIC-15 Content Validation

The evaluation of each item as indicative of pain (Question A) is represented by the CVC in Table 1, sorted by HCPs separately and the overall group view. The assessment of each item as specific for detecting pain in cognitive impairment or whether they were signs that meant other conditions (such as delirium, depression, anxiety, or dementia) (Question B) is presented in Figure 1 (overall group opinion) and Figure 2 (target users’ input).

In general, the target users agreed on 13 items (87%) regarding whether the item was indicative of pain. Shouting and Restlessness were the two non-consensus items considered indicative of pain by the geriatricians who do not work in LTCF but not by the geriatricians and nurses who work in LTCF. Yet, 87% of agreement was seen among the target users regarding the specificity of each item. Differences were seen in shouting, which was considered specific for pain for the nurses who work in LTCF but not for the geriatricians, and the opposite was observed for resisting care.

The non-target users, however, showed various opinions, and agreement among the professionals about the item being indicative of pain was present only for five items (30%). The findings regarding each item’s characteristics of being indicative or specific of pain are presented below according to the domains: facial expression, body movement, and vocalization, with an overall group insight or from the point of view of the target users.

#### 3.2.1. Facial Expressions

Frowning and narrowing the eyes were considered indicative and specific for pain in the overall analysis and for the target users, respectively. Additionally, looking tense also indicated pain in the overall group and among the target users. Over half of the target users agreed that this item was specific for pain; however, the overall group achieved only 49.5% agreement.

Opening the mouth was the item with the lowest CVC: only 25% of the overall group agreed that this was a specific item for pain, and the majority believed it meant dementia. Rising upper lips were not considered indicative of pain by any professional, nor was it considered specific in the overall group view. However, it was deemed specific to pain among the target users.

#### 3.2.2. Body Movement

Freezing, guarding, resisting, and rubbing had acceptable CVC as indicative of pain in the overall group and among target users. Freezing, guarding, and rubbing were accepted by more than 50% of participants among the overall group and the target users. Resisting was not identified as specific to pain among nurses or all participants. On the other hand, geriatricians believed that it could be specific to pain. Restlessness achieved acceptable CVC as indicative of pain among nurses and geriatricians who did not work in LTCF; however, it was considered specific to pain only among 33% of professionals in the overall group. The item restlessness was not considered specific to pain among the target users.

#### 3.2.3. Vocalization

Using pain-related words, groaning, and complaining were considered indicative and specific of pain in the overall group and among target users. Shouting presented an acceptable CVC among nurses working in LTCF and geriatricians who did not work in LTCF. It was considered specific to pain by 50% of nurses in LTCF, but not for the other target users or the overall group. Mumbling was not considered suggestive or specific of pain in any of the participant groups.

### 3.3. Agreement Between the Brazilian and the Dutch Versions of PAIC-15

Table 2 compares the opinions of similar professionals in Brazil (target users) and the Netherlands. There was general agreement regarding the specificity of pain between the nurses and physicians in the Netherlands regarding 13 items (87%), except for *frowning* and *freezing*. In the Brazilian version, the same level of agreement was found among target users, with disagreements related to *resisting* and *shouting*. Brazilian and Dutch professionals generally agreed on ten items (67%), saying that *narrowing*, *raising lips*, *guarding*, *using pain-related words*, *rubbing*, *groaning*, and *complaining* are specific to pain. On the other hand, *opening the mouth*, *restlessness*, and *mumbling* were not considered specific to pain in both countries.

### 3.4. Open-Ended Questions

***Theme 1****: Non-specificity of some items*. Some participants reinforced the lack of specificity regarding pain in PLWD in some of the tool’s items. Considering the difficulty for people with advanced cognitive impairment to express feelings of hunger, fear, cold, discomfort, or insecurity, one of the participants mentioned that *“Some items might represent another discomfort, like an itch, hunger, or lack of interest*.” (P7, Geriatrician LTCF). Participants also mentioned that concomitant signs might be helpful when assessing the items in PLWD: “*Sweat plus a behavior change might be suggestive of pain in patients with dementia*.” (P19, Geriatrician).

***Theme 2****: Influence of the level of cognitive impairment and person-centered care on item interpretation*. As dementia progresses, the inability to understand and articulate the experience of pain can cause professionals, family members, and caregivers to underestimate its presence. One participant highlighted: “*The assessment depends on the severity and comorbidities of dementia. For example, freezing can be related to Parkinson’s disease*” (P30, Geriatrician LTCF). Caregivers familiar with a patient with severe dementia play a crucial role in identifying pain by noticing subtle signs of behavioral changes, which can facilitate the adoption of individualized strategies to identify pain in patients with severe dementia: “*The instrument may be applicable if the healthcare professional is familiar with the patient and knows his routine behavior*.” (P46, Nurse LTCF). Indeed, person-centered approaches facilitate the development of faster and more effective strategies to recognize and respond to the pain needs of patients with severe dementia. However, its implementation on a large scale and in all settings is neither uniform nor straightforward.

***Theme 3****: Cultural differences influence the naming of some items*. Cultural differences usually emerge when tools and instruments are translated and culturally adapted into other languages. In addition to linguistic issues, it is interesting to highlight that the perception of professionals who assist PLWD in different contexts and cultures may include insights not initially considered when the original tool was designed. One of the participants, for example, suggested: “Changing ‘care resisting’ to ‘resisting the manipulation of his/her body,’ because ‘care resisting’ may mean annoyance and not pain” (P63, Nurse). In addition to linguistic issues, the interpretation of some items may also vary in different cultures. Expressions that can be easily understood in a language may not have an exact or sufficiently precise counterpart when translated. One of the participants, for example, reported: “I did not understand the item ‘opened mouth’; was it the process of opening the mouth or keeping it open?” (P56, Geriatrician LTCF). Despite the efforts of research judges, translators, authors of the original version, and clinical experts, it is possible that items are still challenging to understand, and these points may be gaps that may lead to revisions and reevaluations of the properties and descriptions of the original items in the tool.

## 4. Discussion

We undertook the translation, cultural adaptation, and content validation of the PAIC-15 scale in Brazil. We compared the findings with those obtained in translating, adapting, and validating PAIC-15 from English to Dutch [15]. The Brazilian Portuguese version of PAIC-15 showed acceptable content validity between the target users and when compared to Dutch professionals, who have a different reality.

In the present study, only 52% of the participants reported the use of a structured tool to assess pain, corroborating the previous data that shows that pain in older people with impaired cognition is underdiagnosed and undermanaged [8,19,20]. Liao et al. 2023 raised several evidence-based barriers and facilitators to managing pain in persons with dementia [4]. They identified, for instance, the lack of policies, such as the lack of guidelines, inconsistency in the pain assessment tool, and lack of documentation, as barriers. Implementing a structured pain assessment tool is one of the facilitators of improving pain management because it can help the HCP identify discrepancies, prompt recognition of pain or pain-related behaviors, and help less experienced HCPs assess pain. The pain assessment tool needs to address non-verbal manifestations of pain-related behaviors, such as agitation and aggression [21,22]. Since the ones with cognitive impairment are less likely to receive pain treatment, this fact can lead to a troubled environment, increase the risk for other comorbidities, reduce quality of life, and increase the burden on the caregivers [20,23]. The PAIC-15 is a promising multidomain tool with properties to detect these non-verbal behaviors that can mean pain. Therefore, this study aims to make PAIC-15 available, resulting in one more pain assessment tool option for Brazilian HCPs to include in their protocol and care programs for older people with impaired cognition, thus making pain detection and management more inclusive.

Only a few scales to assess pain in PLWD are available and validated in Brazil. One example is the “Pain Assessment in Advanced Dementia” (PAINAD), and another is “Pain Intensity Measures for People with Dementia Portuguese” (PIMDp) [24]. Both PIMDp and PAIC-15 involve assessing facial expression, body movements, and vocalization through a combination of items from existing scales to assess pain. However, while PAIC-15 points to each item individually, the PIMDp combines some items.

Evaluation of the superiority of any pain scale over the others remains to be undertaken. Without the gold standard, it is essential to explore different options, especially a simple one that is well-studied, has good psychometric properties, and is broadly available in various languages, such as PAIC-15.

In the Brazilian version, the target users recognized more items as indicative of pain than the non-target users. The non-geriatrician physicians did not identify any facial expressions indicative of pain. LTCF nurses and geriatricians agreed 100% on all items, and nurses with LTCF geriatricians agreed on 14 (93%) items. Therefore, the target users have a more uniform opinion because they are more familiar with the hidden signs of pain in the setting of cognitive impairment. In LTCFs, Nurses exert continuous work alongside residents, and physicians generally treat acute or chronic conditions in brief moments. These few discrepancies may be related to the pattern of care. In a qualitative study, LTCF nurses were asked about their uncertainty in managing pain in people with impaired cognition and living in LTCF. From the different dimensions raised, the nurses mentioned that they are overexposed to the unmet needs, and they are accustomed to identifying vocal and aggressive behaviors as indicative of or specific to pain. The LTC geriatricians’ care is probably not as close and intense as the nurses’ care is [25,26].

Concerning facial expression, many discrepancies were observed regarding the indication or specificity of pain among the observers and between the Brazilian and Dutch translated versions. Frowning, narrowing, and looking tense were indicative and specific of pain for all the Brazilian target users, but there was no consensus among the Dutch. A possible explanation is that facial expressions are subtle and fleeting and are thus complex to recognize [27]. Some are subjective and cultural differences, like looking tense [10]. Previously, studies have shown discrepancies in interpreting this item between observers from different countries [27,28]. However, facial responses are important and valid in assessing pain, as previously reported [29]. In the present study, one relevant free comment from one responder was that they did not understand what opening mouth means, so a better explanation and examples of the items could be helpful. Therefore, it is possible that less training and cultural differences are involved in the discrepancies observed between the professionals.

Fewer discrepancies were regarding the domain of body movements, except for the item resisting. The Brazilian LTCF nurses had the same opinion as the Dutch clinical experts that resisting was not indicative or specific for pain. However, in the present study, an exciting reflection came from one participant suggesting we could use “resisting the manipulation of the body” instead, as this could be more specific about pain because care-resisting could simply mean that the person was annoyed. Therefore, it is possible that the item reflects pain and needs to be clarified. Guarding and rubbing were unanimous among Brazilian target users, and Dutch opinion was that they were indicative and specific for pain. Guarding was the item with the highest CVC for each Brazilian HCP. In the overall group or separate responses, nearly 90% of professionals in both Nations agreed that guarding is specific for pain.

On the other hand, restlessness was one item that could be indicative of pain for some Brazilian professionals but not for the Dutch. In terms of non-specificity of restlessness, again, there is unanimity in both countries in the overall group opinion or separate analysis. In fact, more anatomical signs are easily recognized as indicative or specific of pain than more subjective items [13].

The vocalization domain presented lower discrepancies compared with the Dutch. The Brazilian target users agreed with the Dutch experts on most items, except for “shouting.” The responses for “Using pain-related words,” “groaning,” and “complaining” are uniform, as the Brazilian overall group and target users and the Dutch agreed the items are suggestive and specific for pain. Of note, “Using pain-related words” was identified for nearly all professionals in Brazil and 100% in the Netherlands as specifically for pain [15].

The PAIC-15 initiative’s principle is to have a universal validation scale that allows data about pain in PLWD to be gathered to implement and control strategies further to optimize pain management. In this sense, the present study moved forward by content validating PAIC-15 in Brazilian Portuguese, with a reasonable multi-professional sample size, involving target users and other professionals who deal with PLWD but are less familiar with the non-verbal pain signs. In Brazil, the Society for Geriatrics and Gerontology is considered relatively new, and there are not enough specialists to look after the aging population and to train the HCPs, so we considered it important to include the non-specialist group who assist this population [30,31]. The discrepancies between these two groups indicate the need for education and training regarding assessing pain in PLWD. Having a structured and validated multidomain tool to assess pain in this neglected population should be part of the policies and procedures for caring for older people [4]. In this sense, PAIC-15 has been considered a potential option for clinical practice, and the reasons for selecting the 15 items were based on several factors, including reliability, validity, and clinical usefulness, as well as the balance between the domains (5 items per domain: vocalization, facial expression, and body) [13,23]. Therefore, as Van Dalen-kok et al., we decided not to dismiss one component at the stage of expert validation due to one poorer characteristic [15]. During the translation from Dutch to English, the same three items were not considered suggestive or specific for pain; however, they were retained in the original version with advice on improving interdisciplinary education on pain assessment, particularly regarding the interpretation of facial expressions. Additionally, it is important to maintain the main structure for the international comparability of PAIC-15. With 80% of the PAIC-15 items validated, the process of translating the English version to Brazilian Portuguese also helped identify potential issues that require further clarification during educational training and additional clinical validation. Furthermore, the PAIC has been the subject of constant research, and our study, in combination with Van Dalen-Kok’s, sheds light on the direction of creating a short version with fewer items.

Regarding the specificity, in a systematic review that included 12 observational studies with 281.969 participants, pain was associated with 13 Behavior and Psychological Symptoms of Dementia (BPSD), including delirium, depression, and anxiety. Therefore, the overlap between BPSD and pain justifies some items not being recognized as specific, but indicative of pain [32].

Finally, the applicability and whether the PAIC-15 would help to assess and manage pain in PLWD need to be further explored.

The present study has some limitations, such as not being tested in clinical practice or evaluating interobserver reliability during pain assessment. The participants were guided to choose only one option relating to the specificity of the item; therefore, they could hide part of their opinion. In addition, it took a lot of work to reach an adequate number of nurse technicians who directly looked after the older people at the LTCF. Non-probabilistic sample recruitment may introduce bias, as participants were not randomly selected. The sample may overrepresent individuals with particular characteristics, such as heightened interest in the study topic or greater motivation to volunteer. However, since the study included professionals from both within and outside LTCFs—such as doctors, nurses, and other healthcare professionals, including geriatric specialists and non-specialists—we sought to minimize the selection bias. In addition, including test–retest reliability in subsequent research phases can strengthen confidence in the instrument’s stability. Moreover, the items “rising upper lips”, “opening the mouth”, and “mumbling” might not be recognized as being directly related to pain in these patients, in the opinion of the Brazilian target users. There is a current need for studies that analyze whether PAIC-15 changes in response to analgesic drugs, reflecting a painful process in these patients.

## 5. Conclusions

After a transparent and rigorous translation and cultural adaptation process, the Brazilian version of PAIC-15 showed acceptable content validity. Brazilian and Dutch professionals agreed on several items. Insights from thematic analysis of linguistic equivalence difficulties might encourage training and education for using the original version and future adaptation methods of the original tool items in larger samples to confirm their construct equivalence and psychometric properties, particularly among items perceived as being poorly specific to pain in PLWD.

## Figures and Tables

**Figure 1 ijerph-22-01324-f001:**
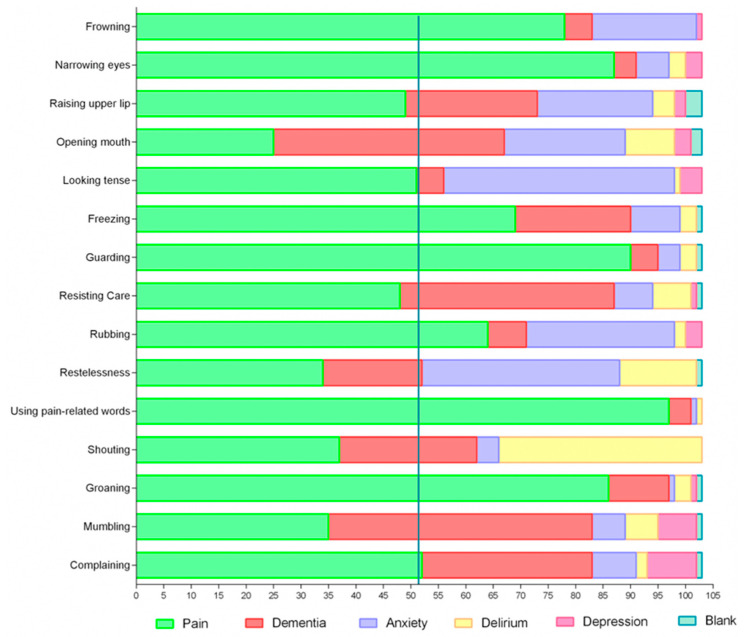
Number of participants who perceived the items of PAIC-15 as specific for pain in cognitive impairment or specific for other conditions such as dementia, anxiety, delirium, or depression. The blue line represents 50% of the participant’s responses.

**Figure 2 ijerph-22-01324-f002:**
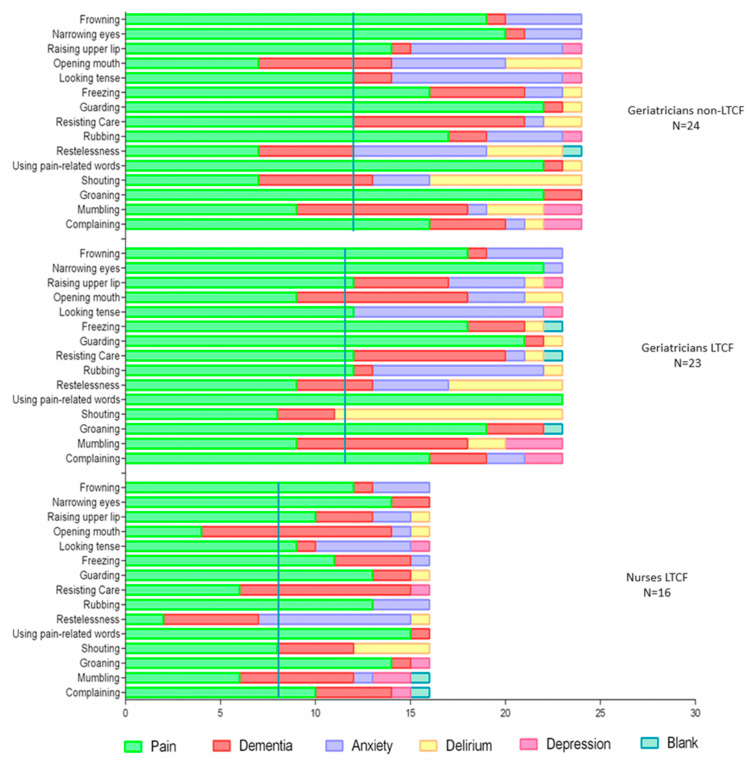
The number of target users (geriatricians and nurses who work at LTCF and geriatricians who work at other settings) who perceived the items of the PAIC-15 as specific for pain in cognitive impairment or specific for other conditions such as dementia, anxiety, delirium, or depression. The blue line represents 50% of the participants’ responses.

**Table 1 ijerph-22-01324-t001:** Content validity coefficient (CVC) scores are used to assess whether the items in the Brazilian version of PAIC-15 suggest the existence of pain.

Items in Brazilian Portuguese	Geriatricians (Other Settings) (*n* = 23)	Geriatricians (LTCF) (*n* = 23)	Other Health Professionals (*n* = 19)	Nurses (LTCF) (*n* = 16)	Physicians (Non-Geriatricians) (*n* = 12)	Nurses (Hospitals) (*n* = 7)	Nursing Technicians (LTCF) (*n* = 2)	Overall Group (*n* = 102)
**FACIAL EXPRESSION**								
“Franzindo a testa” (frowning)	**0.80**	**0.85**	0.78	**0.81**	0.75	**0.82**	0.75	**0.80**
“Apertando os olhos” (narrowing)	**0.83**	**0.84**	**0.80**	**0.86**	0.77	0.79	0.75	**0.82**
“Levantando o lábio superior” (rising upper lips)	0.72	0.71	0.69	0.78	0.69	0.64	0.50	0.71
“Abrindo a boca” (opening the mouth)	0.63	0.68	0.60	0.72	0.63	0.64	0.62	0.65
“Parecendo tenso” (looking tense)	**0.83**	**0.83**	**0.84**	**0.85**	0.77	**0.82**	**0.87**	**0.83**
**BODY MOVEMENTS**								
“Enrijecido ou rígido” (freezing)	**0.83**	**0.97**	**0.80**	**0.94**	0.79	**0.82**	**1.00**	**0.85**
“Postura de proteção” (guarding)	**0.93**	**0.95**	**0.93**	**0.92**	**0.96**	**0.89**	**0.87**	**0.93**
“Resistindo ao cuidado” (Resisting)	**0.86**	**0.82**	0.79	**0.89**	**0.81**	0.75	**0.87**	**0.83**
“Fricção” (Rubbing)	**0.85**	**0.91**	**0.84**	**0.86**	**0.81**	0.79	0.62	**0.85**
“Inquietação” (Restlessness)	**0.80**	0.77	0.71	**0.86**	0.73	**0.82**	**0.87**	0.78
**VOCALIZATION**								
“Usando as palavras relacionadas à dor” (Using pain-related words)	**0.87**	**0.90**	**0.86**	**0.91**	**0.92**	**0.82**	**0.87**	**0.88**
“Gritando” (Shouting)	**0.81**	0.73	0.74	**0.86**	0.73	0.79	**0.87**	0.77
“Gemendo” (Groaning)	**0.82**	**0.89**	**0.83**	**0.83**	**0.81**	**0.86**	0.75	**0.84**
“Resmungando” (Mumbling)	0.67	0.70	0.71	0.77	0.58	0.71	0.75	0.69
“Reclamando” (Complaining)	**0.80**	**0.86**	**0.80**	**0.88**	0.77	0.79	**0.87**	**0.82**

LTCF: Long-term care facility. One “geriatrician working in other settings” did not respond to the questions about whether each item of PAIC-15 appears to be appropriate to “indicate” pain. All the CVCs ≥ 0.80 are highlighted in bold.

**Table 2 ijerph-22-01324-t002:** Comparison of levels of agreement per item among Brazilian and Dutch professionals [12].

Item	Brazilian Nurses LTCF	Brazilian Geriatricians LTCF	Brazilian Geriatricians Non-LTCF	Dutch Nurses LTCF	Dutch Older People’s Care Physicians–LTCF
**FACIAL EXPRESSIONS**					
Frowning	x	x	x		x
Narrowing	x	x	x	x	x
Raising upper lip	x	x	x	x	x
Opening mouth					
Looking tense facial	x	x	x		
**BODY MOVEMENTS**					
Freezing	x	x	x	x	
Guarding	x	x	x	x	x
Resisting		x	x		
Rubbing	x	x	x	x	x
Restlessness					
**VOCALIZATION**					
Using pain-related words	x	x	x	x	x
Shouting	x				
Groaning	x	x	x	x	x
Mumbling					
Complaining	x	x	x	x	x

The “x” and green cells represent that more than 50% of Brazil and the Netherlands participants believed the item was specific for pain. The red cells are highlighted when fewer than 50% of the professionals thought the item was not pain-specific.

## Data Availability

The corresponding author may provide anonymized data for research purposes. The author may also email any additional information or data required to support the study.

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
