# Peer review of "Reducing Barriers for Best Practice in People Living with Dementia: Cross-Cultural Adaptation and Content Validity of the Brazilian Version of the Pain Assessment in Impaired Cognition (PAIC-15) Meta-Tool"

_ijerph, 2025, doi:10.3390/ijerph22091324_

Round 1

Reviewer 1 Report

Comments and Suggestions for Authors

Dear authors,

I find this article interesting because it addresses the assessment of pain in people with severe cognitive impairment who are unable to express their presence.

The title provides adequate information about the topic of the article and would facilitate searching databases.

The introduction broadly addresses the current problems in pain assessment in patients with severe dementia, although I believe that a proper justification should be given for using this questionnaire rather than others for content validation.

The objective is well defined and responds to the topic of study.

The methodology is detailed to allow for reproduction.

The type of sampling and the problems it may cause should be indicated. Selecting the sample using non-probability sampling may lead to a risk of self-selection, which should be indicated in the study limitations.

Wouldn't it have been more appropriate to conduct the study exclusively with professionals specialized in treating pain in these types of patients? I believe the results could be much more representative.

I believe it would have been interesting to conduct an inter-rater reliability analysis to understand the temporal stability of the questionnaire items.

The results are presented in tables and graphs for ease of understanding.

The discussion analyzes the results in detail, comparing them with existing evidence, highlighting aspects specific to the Brazilian population. However, some aspects could be further explored, such as the differences in perception between geriatricians and nurses, or specific practical recommendations for the use of this questionnaire in the Brazilian population with severe dementia.

The limitations derived from the type of sampling and the possibility that the questionnaire items may not be directly related to pain in these patients should be noted. It would be advisable to highlight the need for studies that analyze the items and how they change in response to analgesic drugs to determine whether they reflect a painful process in these patients.

The conclusions are consistent with the results obtained and meet the proposed objective.

The references are appropriate and most are up-to-date.

Kind regards.

Author Response

POINT-BY-POINT RESPONSE TO REVIEWERS

Dear All,

We thank you very much for all the input that helped us to improve the present manuscript. We responded to all comments and suggestions point by point, incorporating all concerns and recommendations made by the reviewers. We hope the new version is suitable for publication in this high-standard journal.

Best regards,

Paula S Azevedo, Juli T. Souza , Patrick Waschholz  and team.

REVIEWER 1

1-“I find this article interesting because it addresses the assessment of pain in people with severe cognitive impairment who are unable to express their presence.

The title provides adequate information about the topic of the article and would facilitate searching databases.

The introduction broadly addresses the current problems in pain assessment in patients with severe dementia, although I believe that a proper justification should be given for using this questionnaire rather than others for content validation.

The objective is well defined and responds to the topic of study.

The methodology is detailed to allow for reproduction. “

            Thank you very much for your time in reviewing our manuscript and for your appreciated input. We are delighted that we were able to convey the message of this study.          The modifications in the previous manuscript, as suggested, and to address your concerns, are highlighted in red in the manuscript with track changes.

            We improved the justification for translating PAIC 15 to Portuguese instead of other questionnaires. (Section: 1. Introduction / Paragraphs: 5 and 6)

 “2. The type of sampling and the problems it may cause should be indicated.

Selecting the sample using non-probability sampling may lead to a risk of self-selection, which should be indicated in the study limitations.

            Thank you for bringing this critical topic to our attention. We acknowledge that utilizing non‑probability sampling introduces potential limitations. Because participants were not randomly selected, the sample may over‑represent individuals with particular characteristics—such as heightened interest in the study topic or greater motivation to volunteer—resulting in self‑selection bias.

            This may limit the representativeness of the sample and the generalizability of findings to the broader population.

            To help mitigate this bias, we also describe our recruitment methods in detail, and we compare participant demographics to known population parameters wherever possible. However, statistical inference to a larger population should be interpreted cautiously.

            First, we included the information: “A non-probabilistic sample was used; however, we utilized virtual-based spaces and a national conference to attempt to reach healthcare professionals involved in the care of older adults.” (Section: 2.2.2. Procedures / Paragraph: 4)

            Second, we have added a paragraph under Limitations to acknowledge the risk and its implications for external validity, as described below in response for your next concern. (Section: 4. Discussion / Paragraph: 13)

  1. I believe it would have been interesting to conduct an inter-rater reliability analysis to understand the temporal stability of the questionnaire items.”

            We appreciate the suggestion regarding inter‑rater reliability and temporal stability testing. While our survey instrument uses self-administered questionnaire items rather than multiple raters, your point is well taken regarding test–retest reliability, which assesses the stability of respondents’ scores over time.

            As this was a study aiming to translate the PAIC-15 to Portuguese, not evaluating psychometric properties of the tool, we have added a statement in Limitations acknowledging the absence of such analyses in the current study, and a plan for including test–retest reliability in subsequent research phases to strengthen confidence in the stability of the instrument. We described this modification below. (Section: 4. Discussion / Paragraph: 13)

4.The limitations derived from the type of sampling and the possibility that the questionnaire items may not be directly related to pain in these patients should be noted. It would be advisable to highlight the need for studies that analyze the items and how they change in response to analgesic drugs to determine whether they reflect a painful process in these patients.

            Thank you for the suggestions. Based on your review, we updated all the limitations. (Section: 4. Discussion / Paragraph: 13)

“The present study has some limitations, such as not being tested in clinical practice or evaluating interobserver reliability during pain assessment. The participants were instructed to select only one option regarding the specificity of the item, which could conceal part of their opinion. Besides, it took a lot of work to recruit an adequate number of nurse technicians who directly cared for the older people at the LTCF. Non-probabilistic sample recruitment may introduce bias, as participants were not randomly selected. The sample may overrepresent individuals with particular characteristics, such as heightened interest in the study topic or greater motivation to volunteer. However, since the study included professionals from both within and outside LTCFs—such as doctors, nurses, and other healthcare professionals, including geriatric specialists and non-specialists—we sought to minimize the selection bias. In addition, including test–retest reliability in subsequent research phases can strengthen confidence in the instrument's stability.  Moreover, the items “rising upper lips”, “opening the mouth”, and “mumbling” might not be recognized as directly related to pain in these patients, in the opinion of the Brazilian target users. There is a current need for studies that analyze whether PAIC-15 changes in response to analgesic drugs, reflecting a painful process in these patients.”

  1. Wouldn't it have been more appropriate to conduct the study exclusively with professionals specialized in treating pain in these types of patients? I believe the results could be much more representative

            Thank you very much for bringing this critical point to our attention. However, the geriatric specialty in Brazil is relatively new, and, commonly, non-specialists care for older people with impaired cognition. Therefore, it is crucial to hear the perspectives of these HCPs. For this reason, we conducted an analysis that combined all the answers and distinguished between the “target population” and non-specialists.  We have already discussed this issue in a previous paper published by our group. So we included this explanation and reference in the manuscript. (Section: 4. Discussion / Paragraph: 10)

 “In Brazil, the Society for Geriatrics and Gerontology is considered relatively new, and there are not enough specialists to look after the aging population and to train the HCPs, so we considered it important to include the non-specialist group who assist this population.”

6.The results are presented in tables and graphs for ease of understanding.

6.The discussion analyzes the results in detail, comparing them with existing evidence, highlighting aspects specific to the Brazilian population. However, some aspects could be further explored, such as the differences in perception between geriatricians and nurses, or specific practical recommendations for using this questionnaire in the Brazilian population with severe dementia.

            Thank you very much for your appreciated suggestion. We updated the discussion to incorporate your suggestion about exploring the differences in perception between nurses and geriatricians. (Section: 4. Discussion / Paragraph: 5)

“In LTCFs, Nurses exert continuous work alongside residents, and physicians generally treat acute or chronic conditions in brief moments. These few discrepancies may be related to the pattern of care. In a qualitative study, LTCF nurses were asked about their uncertainty in managing pain in people with impaired cognition and living in LTCF. From the different dimensions raised, the nurses mentioned that they are overexposed to the unmet needs, and they are accustomed to identifying vocal and aggressive behaviors as indicative of or specific to pain. The LTC geriatricians' care is probably not as close and intense as the nurses' care is.”

            About the practical recommendation for the Brazilian population, we added a paragraph in blue to address the comment of reviewer 2, where we highlight the need for interdisciplinary education activities, in particular to clarify the items with poor CVC, the fact that 80% of the items were validated, but the need for more clinical investigation regarding the usability of the tool to help pain management. (Section: 4. Discussion / Paragraph: 10)

7.The results are presented in tables and graphs for ease of understanding.

8.The conclusions are consistent with the results obtained and meet the proposed objective.

9.The references are appropriate and most are up-to-date.

            Thank you again for all the meaningful consideration that helped us to improve the quality of the present manuscript.

Reviewer 2 Report

Comments and Suggestions for Authors
  • A brief summary 

For adequate pain management for individuals with impaired cognition, this study aimed to translate and culturally adapt the original English version of PAIC-15 for use in Brazil, establish its content validity, and compare the findings with a previous similar study developed in the Netherlands.

  • Strengths

The gap is clearly identified, and this project is valuable and should be done.

The translation method is valid, CVC calculations are also good, and the interpretation of the results is sound.   Tables and Figures are useful, showing clear results. Up to this point, the paper is well organized, informative and well written; however, its usability is questionable.

  • Weaknesses

To develop a measurement tool, items with low CVC scores should be revised or removed. In the Brazilian version of PAIC-15, three items (rising upper lip, opening the mouth, and mumbling) are considered problems. The reason for keeping them is not stated.  Whether this is the final product or still in a progress stage is unclear.  If this is the final product, it cannot be satisfactorily used.  

In the sample section, state inclusion and exclusion criteria.

There are inconsistent sample sizes. On Line 170, the total sample size is 103, but in Table 1, overall group is 102. On Table 1, geriatricians other than LTCF is 23, but in Figure 2, it is 24.

Author Response

POINT-BY-POINT RESPONSE TO REVIEWERS

Dear All,

We thank you very much for all the input that helped us to improve the present manuscript. We responded to all comments and suggestions point by point, incorporating all concerns and recommendations made by the reviewers. We hope the new version is suitable for publication in this high-standard journal.

Best regards,

Paula S Azevedo, Juli T. Souza , Patrick Waschholz  and team.

REVIEWER 2

  • A brief summary

For adequate pain management for individuals with impaired cognition, this study aimed to translate and culturally adapt the original English version of PAIC-15 for use in Brazil, establish its content validity, and compare the findings with a previous similar study developed in the Netherlands.

  • Strengths

The gap is clearly identified, and this project is valuable and should be done.

The translation method is valid, CVC calculations are also good, and the interpretation of the results is sound.   Tables and Figures are useful, showing clear results. Up to this point, the paper is well organized, informative, and well written; however, its usability is questionable.

            Thank you very much for your time in supporting this revision process. We are delighted that you appreciated our manuscript for its organization and informative content. We addressed your concerns to further clarify and enhance its usefulness. The modifications made due to your suggestion are highlighted in blue.

  • Weaknesses

1- To develop a measurement tool, items with low CVC scores should be revised or removed. In the Brazilian version of PAIC-15, three items (rising upper lip, opening the mouth, and mumbling) are considered problems. The reason for keeping them is not stated.  Whether this is the final product or still in a progress stage is unclear.  If this is the final product, it cannot be satisfactorily used.  

            Thank you for bringing this critical issue to our attention. We clarified that it is under research and not the end of history. Indeed, the Dutch group has been working on the PAIC-6, and our study will contribute to the selection of the items.  We also clarified the required validation in clinical practice. However, as the Dutch research group, we decided not to remove the items to maintain the balance between the domains, to avoid losing the psychometric properties previously studied, but to highlight the need for education in understanding the meaning of the items. In this context, It is important to retain these items to ensure the international comparability of the PAIC15, as facial expressions may vary across cultures and settings, and in other environments, these items may still be indicative of pain.  The following paragraph was added in the manuscript. (Section: 4. Discussion / Paragraph: 10)

“In this sense, PAIC-15 has been considered a potential option for clinical practice, and the reasons for selecting the 15 items were based on several factors, including reliability, validity, and clinical usefulness, as well as the balance between the domains (5 items per domain: vocalization, facial expression, and body. Therefore, as Van Dalen-kok et al, we decided not to dismiss an item at the stage of expert validation due to one poorer characteristic. During the translation from Dutch to English, the same three items were not considered suggestive or specific for pain; however, they were retained in the original version with advice on improving interdisciplinary education on pain assessment, particularly regarding the interpretation of facial expressions. It is important to retain these items the international comparability of PAIC-15. With 80% of the PAIC-15 items validated, the process of translating the English version to Brazilian Portuguese also helped identify potential items that require further clarification during educational training and additional clinical validation. In addition, the PAIC has been the subject of constant research, and our study, in combination with Van Dalen-Kok´s, sheds light on the direction of creating a short version with fewer items.”

2- In the sample section, state the inclusion and exclusion criteria.

            We clarified the inclusion and exclusion criteria on the manuscript. (Section: 2. Materials and Methods; Subsection: 2.2.1. Sample / Paragraph: 1)

 “The inclusion criteria were HCPs aged 18 and older with a minimum of experience in pain assessment and management, as well as providing care for people with dementia. The exclusion criteria were that the participant did not complete both parts of the questionnaire.”

3-There are inconsistent sample sizes. On Line 170, the total sample size is 103, but in Table 1, the overall group is 102. On Table 1, geriatricians other than LTCF is 23, but in Figure 2, it is 24.

            Thank you for detecting this mistake. We update the results to inform that one participant responded only to part B and the general characteristics. Therefore, the sample size differs for the question of whether the items are indicative of pain (102) and specifically for pain. (Section: 3. Results / Paragraph: 1)

“One hundred and three HCPs from different Brazilian regions responded to the questionnaire. One participant from the non-LTCF geriatrician group did not answer whether the items were considered indicative of pain, so the sample size for Part A was 102.”

Reviewer 3 Report

Comments and Suggestions for Authors

Azevedo et al. translated the Pain Assessment in Impaired Cognition Scale (PAIC-15) from English into Brazilian Portuguese, largely using the same international study design that Dalen-kok's group had already used to translate the PAIC-15 from English into Dutch (van Dalen-Kok AH et al.. Pain Assessment in Impaired Cognition (PAIC): content validity of the Dutch version of a new and universal tool to measure pain in dementia. Clin Interv Aging. 2017 Dec 22;13:25-34. doi: 10.2147/CIA.S144651. PMID: 29317807; PMCID: PMC5743184. The PAIC-15 is an observational tool specifically designed to assess pain in individuals with impaired cognition and those with dementia, making it to a very important tool in old age psychiatry. It consists of 15 behavioral parameters divided equally across three categories: facial expressions, body movements, and vocalizations. The present study was conducted with the support of 103 healthcare professionals, one group being the target users (nurses and geriatricians in LTCFs or other settings) and the other group including healthcare professionals. Half of all participants had experience in assessing pain. The authors performed a translation, cultural adaptation and a content validation in PwD with severe cognitive impairment. The analyses shows that 13 of the 15 items (87%) were validated as indicative and/or specific to pain, of which 10 items were consistent with the Dutch results (67%). The authors successfully validated the PAIC-15 scale in Brazilian Portuguese and performed a translation, cultural adaptation and a content validation in PwD with severe cognitive impairment. The study rationale and translation follows a very important clinical need in dementia care.

I have very few comments:

  1. As the authors carefully elaborate on the great importance of the scale in dementia care and the clinical necessity of pain assessment in people with dementia I would suggest to add recent literature on the impact of unmet pain treatment in dementia care to the paper, e.g. Shi T, Xu Y, Li Q, Zhu L, Jia H, Qian K, Shi S, Li X, Yin Y, Ding Y. Association between pain and behavioral and psychological symptoms of dementia (BPSD) in older adults with dementia: a systematic review and meta-analysis. BMC Geriatr. 2025 Feb 14;25(1):100. doi: 10.1186/s12877-025-05719-w. PMID: 39953384; PMCID: PMC11829437.
  2. Line 120-127: In the validation process from English to Dutch Dalen-kok et al. studied a group of nurses in LTCF and geriatricians in any setting, whereas Azevedo et al. additionally included other HCP working in long-term care facilities. Why didn’t the group chose to use the same group of HCP as Dalen-kok etal.?

Author Response

POINT-BY-POINT RESPONSE TO REVIEWERS

Dear All,

We thank you very much for all the input that helped us to improve the present manuscript. We responded to all comments and suggestions point by point, incorporating all concerns and recommendations made by the reviewers. We hope the new version is suitable for publication in this high-standard journal.

Best regards,

Paula S Azevedo, Juli T. Souza , Patrick Waschholz  and team.

REVIEWER 3

Azevedo et al. translated the Pain Assessment in Impaired Cognition Scale (PAIC-15) from English into Brazilian Portuguese, largely using the same international study design that Dalen-kok's group had already used to translate the PAIC-15 from English into Dutch (van Dalen-Kok AH et al.. Pain Assessment in Impaired Cognition (PAIC): content validity of the Dutch version of a new and universal tool to measure pain in dementia. Clin Interv Aging. 2017 Dec 22;13:25-34. doi: 10.2147/CIA.S144651. PMID: 29317807; PMCID: PMC5743184. The PAIC-15 is an observational tool specifically designed to assess pain in individuals with impaired cognition and those with dementia, making it to a very important tool in old age psychiatry. It consists of 15 behavioral parameters divided equally across three categories: facial expressions, body movements, and vocalizations. The present study was conducted with the support of 103 healthcare professionals, one group being the target users (nurses and geriatricians in LTCFs or other settings) and the other group including healthcare professionals. Half of all participants had experience in assessing pain. The authors performed a translation, cultural adaptation and a content validation in PwD with severe cognitive impairment. The analyses shows that 13 of the 15 items (87%) were validated as indicative and/or specific to pain, of which 10 items were consistent with the Dutch results (67%). The authors successfully validated the PAIC-15 scale in Brazilian Portuguese and performed a translation, cultural adaptation and a content validation in PwD with severe cognitive impairment. The study rationale and translation follows a very important clinical need in dementia care.

            Thank you very much for your time in supporting this revision process. We are delighted to read your summary and see that the manuscript delivered a clear message to the readers.

            We addressed the manuscript with your suggestions, and the modifications resulting from your comments are highlighted in green

I have very few comments:

  1. As the authors carefully elaborate on the great importance of the scale in dementia care and the clinical necessity of pain assessment in people with dementia I would suggest to add recent literature on the impact of unmet pain treatment in dementia care to the paper, e.g. Shi T, Xu Y, Li Q, Zhu L, Jia H, Qian K, Shi S, Li X, Yin Y, Ding Y. Association between pain and behavioral and psychological symptoms of dementia (BPSD) in older adults with dementia: a systematic review and meta-analysis. BMC Geriatr. 2025 Feb 14;25(1):100. doi: 10.1186/s12877-025-05719-w. PMID: 39953384; PMCID: PMC11829437.

            Thank you very much for suggesting this relevant reference. We have included it to justify the items that did not met 50% of agreement regarding specificity. They probably reflects the overlap between pain and BPSD.

            We added the paragraph below in the manuscript.  (Section: 4. Discussion / Paragraph: 11)

“Regarding the specificity, in a systematic review that included 12 observational studies with 281.969 participants, pain was associated with 13 Behavior and Psychological Symptoms of Dementia (BPSD), including delirium, depression, and anxiety. Therefore, the overlap between BPSD and pain justifies some items not being recognized as specific, but indicative of pain.”

  1. Line 120-127: In the validation process from English to Dutch Dalen-kok et al. studied a group of nurses in LTCF and geriatricians in any setting, whereas Azevedo et al. additionally included other HCP working in long-term care facilities. Why didn’t the group chose to use the same group of HCP as Dalen-kok metal.?

            Thank you for this critical question. As we responded to reviewer 1, differently from the Netherlands and other developed countries,  in Brazil, it is very common to have the non-specialist caring for older people with dementia. We consider a strength point to include the evaluation of the target users and other HCPs.

Round 2

Reviewer 1 Report

Comments and Suggestions for Authors

Dear Authors,

I consider that the manuscript has been sufficiently improved and I have no additional comments or suggestions.

Kind regards.